# Factors Correlated with Body Image Dissatisfaction in Children and Adolescents Diagnosed with HIV: A Cross-Sectional Study

**DOI:** 10.3390/ijerph192114197

**Published:** 2022-10-30

**Authors:** Suellem Zanlorenci, Michele Caroline de Souza, Cilene Rebolho Martins, Luiz Rodrigo Augustemak de Lima, Diego Augusto Santos Silva

**Affiliations:** 1Sports Center, Research Center in Kinanthropometry & Human Performance, Federal University of Santa Catarina, Florianópolis, Santa Catarina 88040-900, Brazil; 2Centenario Methodist College, Santa Maria, Rio Grande do Sul 97050-180, Brazil; 3Department of Physical Education, Federal University of Alagoas, Maceió, Alagoas 57072-970, Brazil; 4Faculty of Health Sciences, Universidad Autónoma de Chile, Providencia 7500912, Chile

**Keywords:** body image dissatisfaction, body composition, child health, adolescent health, HIV

## Abstract

(1) Background: The adverse effects of continuous use of antiretroviral therapy can generate dissatisfaction with body image in children and adolescents diagnosed with Human Immunodeficiency Virus infection (HIV) infection. The aim of this study was to verify the prevalence and factors associated with body image dissatisfaction (BI) in children and adolescents diagnosed with HIV infection; (2) This is a cross-sectional study with 60 children and adolescents (32 females; 28 males) aged 8–15 years diagnosed with HIV, carried out in Florianópolis, Brazil. Information on BI was collected through a silhouette scale, as well as sociodemographic information, biological maturation, lifestyle, self-esteem, adherence to antiretroviral treatment and aspects of HIV infection; (3) Results: The results indicated that 53.13% of female patients and 53.57% of male patients were dissatisfied with their BI. Lower subscapular skinfold and higher calf skinfold values were associated with BI dissatisfaction in females. Pre-pubertal maturation stage, higher economic level, lower concentrations of CD4+ lymphocytes, lower viral load, lower level of physical activity and longer time in front of the computer and/or video game were associated with BI dissatisfaction in males; (4) Conclusions: BI dissatisfaction was observed in more than half of patients and morphological, economic, behavioral and HIV infection aspects were associated with BI dissatisfaction.

## 1. Introduction

The Human Immunodeficiency Virus (HIV), which causes the Acquired Immunodeficiency Syndrome (AIDS), has raised complex issues and brought to the scientific debate issues such as sexuality, death, physical deformation, discrimination, stigma, marginalization and vulnerability, transforming and reformulating feelings and values rooted in society [1]. During the discovery of the first cases of HIV, there were high mortality rates; however, with the introduction of combined antiretroviral therapy (ART) in the 1990s, the characterization of a fatal disease was changed to a controllable chronic disease [1,2].

Body image consists of a multidimensional construct composed of representations about body size and appearance and emotional responses associated with the degree of satisfaction raised by these perceptions [3]. In this sense, two specific aspects of body image can be distinguished: body size and shape perceptions and feelings towards the body and its regions [4]. Other studies with individuals diagnosed with HIV infection in adulthood have shown that these patients would either decrease their life expectancy or accept the increased risk of mortality in exchange for maintaining their health status free of disease-related bodily changes [5,6,7], an aspect that may influence antiretroviral treatment.

The use of ART in children and adolescents diagnosed with HIV infection aims to reduce viral replication to undetectable levels, rebuild the immune function, reduce the risk of resistance to antiretrovirals and promote lower toxicity [1]. However, the adverse effects on physical appearance are significant and the presence of body image dissatisfaction in children and adolescents diagnosed with HIV infection and associated factors, such as older age, higher body weight, low self-esteem, higher body fat, higher body mass index (BMI) and low level of physical activity [8,9,10], which added to biological, cognitive, emotional and social changes of childhood and adolescence [4] can lead to more serious psychological and behavioral problems.

The decrease in CD4 and CD8 lymphocytes and increase in viral load, that is, the clinical parameters of the disease, can have significant effects on the physical appearance of individuals diagnosed with HIV infection [11], which can directly affect self-esteem, increasing body image dissatisfaction. From the patient’s perspective, adherence to ART reduces the risk of virologic failure, increases survival, reduces the risk of disease progression and the development of resistant viral strains, in addition to improving quality of life [11]. On the other hand, side effects such as increase or loss of fat in certain body regions can modify social relationships and generate greater body image dissatisfaction [12].

However, the association between body image and clinical parameters in relation to HIV (CD4, CD8 lymphocytes and viral load) has not yet been explored in children and adolescents. In HIV+ adults, the signs and symptoms of HIV diagnosis (CD4+, CD8+ and viral load) have been reported to negatively affect body image [13,14,15].

This investigation is important, since HIV infection is visibly expressed in the body through bodily changes that may occur due to the symptoms of the disease (thinness) or also through the use of ART (body fat accumulation or loss), which can modify social relationships and generate greater body image dissatisfaction in children and adolescents diagnosed with HIV infection [14]. Furthermore, verifying factors related to body image dissatisfaction will be relevant to educators and health professionals, so that they have subsidies to develop strategies for prevention and treatment interventions for such disorders. In addition, understanding these correlated factors will facilitate social inclusion actions for these young people, reducing discrimination and stigma associated with the disease. Therefore, the present study aimed to verify the prevalence of body image dissatisfaction and factors related to body image in children and adolescents diagnosed with HIV infection.

## 2. Materials and Methods

### 2.1. Search Features

This is a cross-sectional study, part of the “Saúde PositHIVa” study, carried out in the city of Florianópolis, Santa Catarina, Brazil, from 2015 to 2016. The study complied with all ethical procedures, with approval by the Research Ethics Committee from the Federal University of Santa Catarina (protocol No. 1.410.144) and by the Ethics Committee from the Joana de Gusmão Children’s Hospital (HIJG) (protocol No. 850.0777). The Free and Informed Consent Form (FICF) was signed by parents or legal guardians of children and adolescents and the Assent Form by children and adolescents. All the methodological details of the macro-project can be consulted in previously published studies [16,17].

### 2.2. Population and Sample

Children and adolescents diagnosed with HIV infection (vertical transmission) participated in the study, being recruited at the outpatient clinic of a regional HIV reference center, located in the city of Florianópolis, Brazil. In 2015, of children and adolescents diagnosed with HIV treated at the hospital, 83 were eligible for the present study. Of these, 14 were excluded because they did not meet the eligibility criteria or refused to participate in the research, four withdrew from participating during the study and five did not participate in all data collection steps necessary for the statistical analysis (Figure 1).

Sample size was calculated a posteriori, taking into account type I error (α = 0.05) and type II error (β = 0.95) to identify factors associated with body image dissatisfaction with mean effect size (0.20) [18]. All calculations were performed using the G*Power^®^ software version 3.1.9.2 (Universitat Dusselfodorf, Düsseldorf, Germany) and then 30% was added for losses and refusals. Thus, for simple and multiple linear regression analysis, the sample of 56 children and adolescents allowed finding associations with mean effect size of 0.20.

### 2.3. Eligibility Criteria

Inclusion criteria were: (1) to present record of HIV infection by vertical transmission in the medical record; (2) to be 8–15 years old; (3) to present clinical and laboratory information in the medical record; (4) to have ability to stand and communicate; (5) to answer the questionnaire related to body image, as well as questions related to the independent variables of the present study: age, sexual maturation, economic level, adherence to ART, CD4 and CD8 lymphocytes, viral load, physical activity, sedentary behavior based on screen time, presence of lipodystrophy (lipoatrophy), body mass index, body fat percentage, skinfolds (triceps, subscapular, abdomen and calf) and self-esteem.

Exclusion criteria were: (1) motor impairment or contraindication for intense exercise; (2) speech, hearing and/or cognition impairment; (3) diseases that change body composition, with the exception of those related to HIV infection and (4) do not present the Assent Form and the Free and Informed Consent Form signed by parents/caregivers (age < 18 years) or by themselves (age ≥ 18 years).

### 2.4. Study Variables

#### 2.4.1. Dependent Variable

Body image perception was assessed using the silhouette scale adapted by Childress et al. [19], based on figures of silhouettes developed by Stunkard and Sorensen [20], validated for the evaluation of Brazilian children and adolescents [21,22]. Children and adolescents indicated, among the eight possible figures for each sex, the silhouette that they considered to represent their current physical shape and the silhouette that represented the way they would like to look [22]. The difference between the current silhouette and the ideal silhouette (score = CS − IS) was calculated, considering variation from −7 to +7 for the body image score. The higher the positive or negative score, the greater the discrepancy between CS and IS, that is, the greater the body image dissatisfaction. To estimate the prevalence of those satisfied and dissatisfied with their body image, individuals were categorized as follows: satisfied with body image (score = zero), would like to increase body weight (negative scores) and would like to reduce body weight (positive scores) [23]. For association analyses, the body image score (without categorization) was considered. Such an analysis strategy has already been used with adolescents [24].

#### 2.4.2. Independent Variables

The independent variables of this study were: age, sexual maturation, economic level, ART, CD4+ and CD8+ lymphocytes, HIV RNA viral load, physical activity, sedentary behavior based on screen time, lipodystrophy (lipoatrophy), body mass index, body fat percentage, skinfolds (triceps, subscapular, abdominal and calf) and self-esteem.

Age was continuously collected and categorized into children (8–10 years) and adolescents (11–15 years). Sexual maturation was analyzed through self-assessment (figures representing stages of pubic hair development) [25,26] according to Tanner’s criteria [27], categorized as follows: stage 1 corresponding to the infantile phase—pre-pubertal; stages 2, 3 and 4 characterize the pubertal period; and stage 5 to the post-pubertal phase, adult phase [27]. Pubertal and post-pubertal categories were grouped due to the low number of subjects in the post-pubertal category (*n* = 2).

Economic level was collected through questionnaire that assesses the purchasing power of families [28]. The classification is based on the amount of material goods and varies, in descending order of purchasing power from A to E. In the present study, purchasing power was categorized into “higher purchasing power” (categories “A” and “B”) and “lower purchasing power” (categories “C”; “D” and “E”).

Information about adherence to treatment was collected through the question “Do you take the medications prescribed by the “Hospital Dia” doctors—HIJG?”, to check adherence to the prescribed medication, answers were categorized into yes and no [29]. Information on HIV infection was obtained from the analysis of medical records, from which information regarding CD4+, CD8+ lymphocyte counts (used continuously) and viral load (log) was collected.

Level of physical activity was collected through the following question: “During the last seven days, on how many days were you physically active for at least 60 min a day?”. Responses were categorized as “physically active” when active for seven days and “not physically active” when active less than seven days a week. This classification was established according to literature recommendations [30].

The time spent on different screens (television [TV], computer and video game) was collected through the question: “How many hours per day of the week do you watch (or use) these devices?” Each of the different questions in relation to the different screen devices (TV, computer and video game) were categorized into individuals who had time greater than or equal to two hours, which were considered to have sedentary behavior and those with time less than two hours who did not present such risk behavior [29,31].

Body mass was measured with Tanita^®^ portable digital scale (BF683W, Arlington Heights, IL, USA), with capacity of 150 kg. For height, Altura Exata^®^ stadiometer (Belo Horizonte, Brazil) with capacity of 213 cm was used. For all anthropometric measurements, the recommendations proposed by the International Society for the Advancement of Kinanthropometry (ISAK) [32] were used. Based on the above, body mass index (BMI) was calculated. To assess weight status, the WHO Growth Curves in relation to BMI/age for children and adolescents were used and categorized as normal weight, thinness and overweight according to WHO cutoff points [33].

Body fat was evaluated using the air displacement plethysmography method, which determines body volume, body density and, consequently, total body fat. The validity for this measure in children and adolescents has already been demonstrated [34]. In air displacement plethysmography, an equation is used [35], and estimates of this equation appear to demonstrate low standard errors of estimate. The BOD-POD equipment (Life Measurement Inc. Concord, CA, USA) was used. Subjects were evaluated with appropriate clothing, and the use of a swimming cap was requested to minimize the effects of hair on the air displacement analysis. During the measurement procedure, children and adolescents remained seated with an upright posture and normal breathing, hands resting on the thigh and feet touching the floor of the device. Three 50-s tests were performed, totaling approximately four minutes of procedure duration. Based on the air displacement plethysmography results, the body fat percentage variable was considered for the present study.

Skinfold thickness was measured using Cescorf^®^ skinfold caliper, (Equipamentos Esportivos Ltd.a., Porto Alegre, Rio Grande do Sul, Brazil), measuring unit of 0.1 mm. For all skinfolds, the recommendations proposed by the International Society for the Advancement of Kinanthropometry (ISAK) [32] were used. Triceps, subscapular, abdominal and calf skinfolds were considered in the present study. For each skinfold, three measurements were performed in a circuit and then the mean for each skinfold was calculated.

Lipoatrophy was diagnosed through changes in body fat distribution through triceps skinfold rated below the 5th percentile [36]. Triceps skinfold was measured to assess changes in body fat distribution [37]. Then, results were categorized into present and absent lipoatrophy.

Self-esteem was assessed using the Rosenberg self-esteem scale [38], which was validated and adapted for Brazilian elementary, high school and higher education students aged 10–30 years [39]. It is a Likert scale with ten closed questions, with subsequent response options: strongly disagree, disagree, agree and strongly agree, with scores ranging from one to four points, respectively, assigned to each item. In questions 3, 5, 8, 9 and 10, answer scores are inverted to calculate the sum of points. The total score of the scale ranges from 10 to 40 points, and the higher the score, the higher the self-esteem. The scale showed adequate internal consistency, with the score of 0.90 for the Cronbach’s alpha coefficient [39]. In general, the instrument assesses the attitude and the positive or negative feeling about oneself and includes questions of personal satisfaction, self-depreciation, perception of qualities, competence, pride, self-worth, respect and feelings of failure [39].

### 2.5. Statistical Analysis

Initially, descriptive analysis was performed through the distribution of means, standard deviation and frequencies. Histograms, kurtosis and asymmetry were used to verify data normality (between −2 and +2). All variables showed normal distribution. The t-student test for independent samples was used to verify difference between the current silhouette, ideal silhouette and body image score according to independent variables. One-way Analysis of Variance (ANOVA) was used to verify differences between current silhouette, ideal silhouette and body image score according to BMI categories. Then, Pearson’s linear correlation was performed between the body image score and continuous variables (age, CD4 and CD8 lymphocytes, viral load, BMI, body fat, skinfolds [subscapular, triceps, abdominal and calf] and self-esteem) and Spearman’s linear correlation between body image score and categorical variables (sexual maturation, economic status, use of ART, physical activity, sedentary behavior based on screen time, and lipoatrophy). Simple linear regression was used to verify association between body image score and independent variables. Variables for the multiple linear regression model were selected from the stepwise variable selection criterion, considering the AIC criterion. Subsequently, the diagnosis of multicollinearity (VIF) was performed and only variables with VIF < 10 remained in the final model [40,41].

The gender variable was used for stratification, as evidence indicates that body image disorders differ between females and males [42]. Statistical tests were performed using the R Commander software (Bell Laboratories, University of Manchester, Manchester, UK), version 4.0.2 at 5% significance level.

## 3. Results

A total of 60 children and adolescents of both sexes aged 8–15 years participated in the research, and of the total sample, 53.33% were dissatisfied with their body image (would like to increase body weight [28.33%] and would like to decrease body weight [25%]). In addition, 53.13% (*n* = 17) of female children and adolescents were dissatisfied with their body image (31.25% [*n* = 10] would like to increase body weight and 21.88% [*n* = 7] would like to reduce weight body). Additionally, 53.57% (*n* = 15) of male children and adolescents were dissatisfied with their body image (25% [*n* = 7] would like to increase body weight and 28.57% [*n* = 8] would like to reduce body weight). The other sample characteristics are described in Table 1 and Table 2.

Female children and adolescents with screen time based on computer and/or video game use on weekends of less than two hours had larger current silhouette (2.07 ± 1.35) compared to those with more than or two hours of screen time (1.20 ± 0.44; *p* = 0.015). Furthermore, female children and adolescents classified as overweight had larger ideal silhouette (3.20 ± 1.30) compared to those classified as eutrophic (1.64 ± 0.90) (*p* = 0.007) (Appendix A). No significant differences were found when comparing the current silhouette, ideal silhouette and body image score in male children and adolescents (Appendix A).

The body image score showed significant negative correlation with age (r: −0.462; *p* = 0.013), that is, as age decreased, body image dissatisfaction increased in male children and adolescents (Appendix A).

No significant associations were found between body image score and independent variables in females for simple linear regression. For multiple linear regression, variables age, sexual maturation, use of ART, CD4 lymphocytes, viral load, lipodystrophy (lipoatrophy), subscapular skinfold, calf skinfold and self-esteem remained in the model. In the final model, body image score was inversely associated with subscapular skinfold (β: −0.236; 95%CI: −0.457; −0.016), that is, female children and adolescents with lower subscapular skinfold values were dissatisfied with their body image. In addition, body image score was directly associated with calf skinfold (β: 0.231; 95%CI: 0.001; 0.462), in which female children and adolescents with higher calf skinfold values were more dissatisfied with their body image. The final model explained 52% of the body image score variability in female children and adolescents diagnosed with HIV infection (Table 3).

For the simple linear regression analysis, body image score was inversely associated with sexual maturation (β: −1.348; 95%CI: −2.663; −0.034), that is, pre-pubertal male children and adolescents were more dissatisfied with their body image (Table 4).

In the adjusted model, body image score was inversely associated with sexual maturation (β: −1.595; 95%CI: −2.615; −0.575), that is, pre-pubertal male children and adolescents were more dissatisfied with their image. In addition, male children and adolescents of higher economic status, lower concentration of CD4+ lymphocytes, lower viral load, longer computer and/or video game time on weekdays, lower levels of physical activity and longer computer and/or video game time on weekends were more dissatisfied with their body image. The final model explained 72.4% of body image score variability in male children and adolescents diagnosed with HIV infection (Table 4).

## 4. Discussion

The present study aimed to verify the prevalence of body image dissatisfaction and factors related to body image in children and adolescents diagnosed with HIV infection. The main findings of the present study were: (1) lower subscapular skinfold values and higher calf skinfold values were associated with body image dissatisfaction in female children and adolescents; (2) pre-pubertal maturation stage, higher economic level, lower amounts of CD4 lymphocytes, lower viral load values, lower level of physical activity and longer time in front of the computer and/or video game on weekdays and weekends were associated with body image dissatisfaction in male children and adolescents.

This study found that body image score was directly associated with calf skinfold, that is, female children and adolescents diagnosed with HIV infection with higher calf skinfold values were more dissatisfied with their body image, as presented greater discrepancy between current and ideal silhouette. This may be justified, as female children and adolescents at pubertal or post-pubertal stages show changes in body composition, caused by increased levels of estradiol that can lead to greater fat accumulation in peripheral regions of the body, such as legs [43], which can generate greater body image dissatisfaction.

In contrast, female children and adolescents diagnosed with HIV infection with lower subscapular skinfold values were more dissatisfied with their body image. These discrepancies in relation to associations of skinfolds (calf and subscapular) with body image dissatisfaction reported in the present study may also be related to the fact that body image was not analyzed in a categorized manner with regard to statistical analysis, according to other studies with children and adolescents diagnosed with HIV infection (satisfied, would like to increase body weight and would like to reduce body weight) [9,10]. Such an analysis could not be performed in the present study, as the number of children and adolescents diagnosed with HIV infection (*n* = 60) participating in the study did not guarantee sufficient statistical power for the association analysis, and possible inferences would be limited.

The present study found that male children and adolescents in the pre-pubertal stage of sexual maturation were dissatisfied with their body image. These findings can be justified by the fact that puberty is characterized by the most abrupt pubertal development, and it is in this period that body changes and differences in body composition between sexes begin to appear [44]. In addition, the ideal of beauty imposed by society for male children and adolescents is a larger and more muscular body [12,42,45,46,47], that is, pre-pubertal children and adolescents tend to compare themselves with their peers of the same sex. In this sense, male children and adolescents diagnosed with HIV infection at a more advanced maturation stage (pubertal) have a more developed body in relation to the physical aspects of puberty, that is, greater muscle mass gain compared to pre-pubertal ones. This fact may explain, at least in part, body image dissatisfaction in children and adolescents in the pre-pubertal maturational stage.

This study found that male children and adolescents diagnosed with HIV infection of higher economic status were more dissatisfied with their body image. The culture and environment in which the individual is inserted establish standards and expectations regarding appearance and which characteristics are most valued in each individual [48]. That is, children and adolescents are increasingly vulnerable to pressure to fit in and feel part of the social environment to which they belong [49]. Body image satisfaction in this age group (children and adolescents) is largely influenced by the media and the social environment in which they live, directly affecting body image [50]. That is, the economic level in which the individual is inserted is influential, as different cultural contexts have different expectations regarding body image and how much this ideal of beauty is internalized within each child and adolescent, and being acceptable in society is a critical factor in determining body image satisfaction [51,52,53,54,55].

Male children and adolescents diagnosed with HIV infection with lower concentration of CD4+ lymphocytes and lower viral load were more dissatisfied with their body image. This finding can be justified by the stigma related to the disease itself. There have been many advances in public policies related to HIV, but from the perspective of stigma related to the disease, little has changed [56]. The fact that HIV is associated with sexuality, certain social groups, moral values, is a fact that, in addition to imminent death (before ART distribution), facilitated the emergence of stigma related to the disease [57]. Associated with all these factors, fear of prejudice can bring about significant changes in the lifestyle and relationships of these children and adolescents, which can generate, in addition to other disorders related to mental health, body image dissatisfaction [13].

The present study found that male children and adolescents who spent longer time in front of the computer and/or video game on weekdays and weekends were more dissatisfied with their body image. Studies have indicated that sedentary behavior based on screen time may be replacing the time devoted to physical activities, which have higher energy expenditure, that is, children and adolescents have opted for less active activities such as, for example, the use of television, computers and video games [58,59,60]. Prolonged use of screens can generate several problems related to mental health, including aspects inherent to social isolation, exposure to unrealistic ideals of beauty, unhealthy social comparisons, reduced sleep, cyberbullying and body image dissatisfaction [61,62]. In addition, the considerable increase in internet access and video games facilitate access to what the media considers the ideal body [63], which can generate greater body image dissatisfaction. In addition, exposure to the “perfect body culture”, imposed by the media, influences body image perception, since it offers an ideal standard of beauty that in most cases, cannot be reached [64].

The present study also identified that male adolescents diagnosed with HIV infection, with lower levels of physical activity were more dissatisfied with their body image. The modernization process increasingly contributes to a less physically active lifestyle and to increase in the consumption of high-calorie foods, which results in higher prevalence of overweight and obesity in children and adolescents [58,59,60]. In this sense, inadequate lifestyle does not only lead to physical health problems, but also to mental health problems, increasing the occurrence of symptoms related to stress, anxiety, depression, eating disorders and disorders in relation to body image [65,66].

The present study found no association between body image dissatisfaction and self-esteem, viral load, CD4 lymphocytes, use of ART, maturation and age in female children and adolescents diagnosed with HIV infection, demonstrating that although the literature reports that these variables can be associated with body image [13], in the present study, these variables did not demonstrate power of association for female children and adolescents. The findings of the present study may be related to the fact that children and adolescents rely more on body adiposity indicators for body image construction, since it is more visible to them that skinfolds are indicators that present greater difficulty of body satisfaction perception [67,68,69].

The present study also found no associations between body image dissatisfaction and body fat percentage, abdominal skinfold thickness and self-esteem in male children and adolescents diagnosed with HIV infection. These results suggest that these male children and adolescents in the population under study do not rely on adiposity indicators (body fat percentage and skinfold thickness) to determine their body image construction, unlike females [67,68,69]. An original study with children and adolescents of both sexes diagnosed with HIV infection reported association between low self-esteem and body image dissatisfaction [8]. This discrepancy between results can be explained by the variety of instruments used to assess body image and self-esteem [55,70]. These different assessment methods can be explained by the increase in scientific production related to body image and the need for instruments designed for certain age groups [71]. In addition, cultural differences in each country can influence the results obtained in each study [52]. It is inevitable that each individual internalizes a set of beliefs, attitudes, values and behaviors, which are transmitted from generation to generation and common to all individuals inserted in a given culture [51].

The results shown in the present study must be interpreted taking into account that the cross-sectional nature of the study makes it impossible to determine cause and effect relationships between variables. In addition, the present study could not identify whether factors correlated with body image dissatisfaction were due to thinness or excess weight, which is another limitation. This categorized analysis could not be performed in the present study, as the number of children and adolescents diagnosed with HIV infection (n = 60) did not guarantee sufficient statistical power for association analyses and inferences would be limited. Furthermore, instruments for measuring independent variables, sedentary behavior and physical activity were based on self-reports. In other words, these measures do not ensure adequate understanding of the proposed questions [72]; however, researchers were properly trained so that they could explain any of these questions in case of doubts on the part of participating children and adolescents. Additionally, the fact that the questionnaire used to assess self-esteem does not have validation for children under 10 years of age can also be considered a study limitation, but at the time of data collection, if the child could not answer a certain question, researchers offered the necessary assistance so that all questions were correctly answered. Despite the study limitations, the presentation of data referring to factors associated with body image score can be considered as a strength of the present study, which can serve as parameter for investigations with populations similar to that of the present study. In addition, unlike other studies on this population [9,10], body fat was assessed by means of air displacement plethysmography, and its validity for children and adolescents has already been evidenced [34].

Educators must understand the universe of students in order to prevent the risks that childhood and adolescence present. [73]. The Physical Education teacher who teaches children has an important role in the process of building body image, as Physical Education is an area of knowledge that teaches aspects of corporeity and health promotion for children and adolescents. These results also allow the Physical Education teacher, when carrying out an assessment of the students’ body image, to follow their evolution in relation to this theme during the school years. Educators can organize lectures, debates, games and theaters among students to work on the topic of body image and highlight the importance of respect among students [73].

## 5. Conclusions

In conclusion, the results of the present study show the presence of body image dissatisfaction and factors associated with body image dissatisfaction that are different between sexes in children and adolescents diagnosed with HIV infection. In female children and adolescents diagnosed with HIV infection, lower subscapular skinfold values and higher calf skinfold values were associated with body image dissatisfaction, while in male children and adolescents diagnosed with HIV infection, pre-pubertal maturation stage, higher economic level, lower amounts of CD4 lymphocytes, lower viral load values, lower level of physical activity and longer time in front of the computer and/or video games on weekdays and weekends were associated with body image dissatisfaction.

## Figures and Tables

**Figure 1 ijerph-19-14197-f001:**
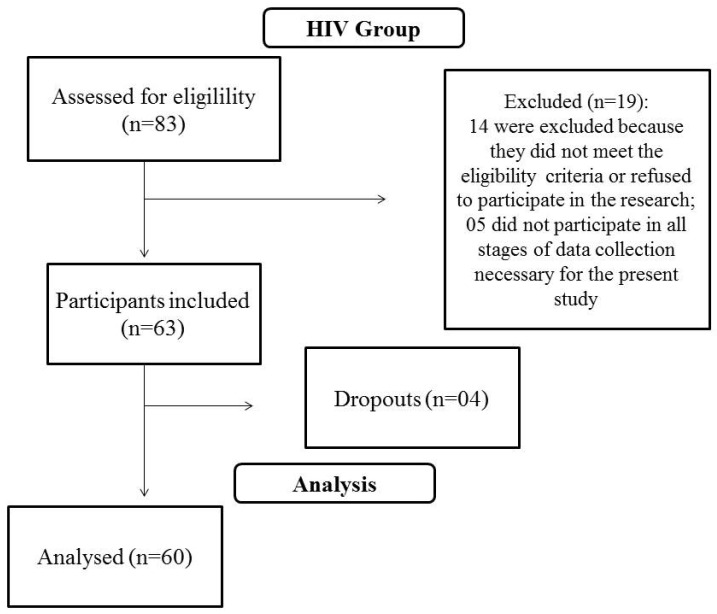
Flowchart of the sampling process of this study.

**Table 1 ijerph-19-14197-t001:** Characteristics and differences between the sexes of the investigated sample of children and adolescents diagnosed with HIV infection (*n* = 60). Florianópolis—SC, 2015/2016.

	Total *n* (%) 60 (100)	Female *n* (%) 32 (53.33)	Male *n* (%) 28 (46.67)	*p*
**Body Image**				0.789
Satisfied	28 (46.67)	15 (46.87)	13 (46.43)	
I would like to increase body weight	17 (28.33)	10 (31.25)	7 (25.00)	
I would like to reduce body weight	15 (25.00)	7 (21.88)	8 (28.57)	
**Sexual maturation**				0.836
Prepubescent	11 (18.33)	5 (15.63)	6 (21.43)	
Pubescent	49 (81.67)	27 (84.38)	22 (78.57)	
**Economic Level**				0.307
High purchasing power	7 (11.67)	5 (15.63)	2 (7.14)	
Low purchasing power	53 (88.33)	27 (84.37)	26 (92.86)	
**ART use**				0.105
Yes	50 (83.33)	29 (90.63)	21 (75.00)	
No	10 (16.67)	3 (9.37)	7 (25.00)	
**Physical activity**				0.142
Physically Active	18 (30.00)	7 (21.87)	11 (39.29)	
Little Physically Active	42 (70.00)	25 (78.13)	17 (60.71)	
**Television Time (week)**				0.131
≥2 h	14 (23.33)	5 (15.63)	9 (31.14)	
<2 h	46 (76.67)	27 (84.37)	19 (67.87)	
**Television time (weekend)**				0.382
≥2 h	25 (41.67)	15 (46.87)	10 (35.71)	
<2 h	35 (58.33)	17 (53.13)	18 (64.29)	
**Computer and Video Game Time (week)**				0.083
≥2 h	39 (65.00)	24 (75.00)	15 (53.57)	
<2 h	21 (35.00)	8 (25.00)	13 (46.43)	
**Computer and Video Game Time** **(weekend)**				**0.009 ***
≥2 h	42 (70.00)	27 (84.37)	15 (53.57)	
<2 h	18 (30.00)	5 (15.63)	13 (46.43)	
**Lipoatrophy**				**0.008 ***
Present	7 (11.67)	7 (21.88)	0 (0.00)	
Absent	53 (88.33)	25 (78.13)	28 (100.00)	
**Body mass index**				0.731
Eutrophy	47 (78.33)	25 (78.12)	22 (78.58)	
Thinness	5 (8.33)	2 (6.25)	3 (10.71)	
Overweight	8 (13.33)	5 (15.63)	3 (10.71)	

Frequency distributions, *n*: number of frequencies, * and bold: *p* < 0.05.

**Table 2 ijerph-19-14197-t002:** Characteristics and differences between the sexes of the investigated sample of children and adolescents diagnosed with HIV infection (*n* = 60). Florianópolis—SC, 2015/2016.

	Total (*n* = 60)	Female (*n* = 32)	Male (*n* = 28)	*p*
Mean (±S.D)				
**Age**	11.93 (6.99)	11.90 (1.94)	11.96 (2.11)	0.913
**CD4 lymphocytes (cells·mm^−^³)**	30.46 (8.97)	30.67 (9.02)	30.21 (9.06)	0.846
**CD8 lymphocytes (cells·mm^−^³)**	1170.10 (545.61)	1174.15 (519.45)	1165.46 (583.67)	0.952
**Viral Load (log)**	2.16 (0.94)	2.17 (1.01)	2.15 (0.89)	0.941
**BMI (kg·m^−2^)**	17.83 (2.71)	18.18 (2.96)	17.44 (2.38)	0.280
**Body fat (%)**	13.77 (6.99)	12.68 (6.59)	15.01 (7.35)	0.205
**TSF (mm)**	9.48 (3.32)	9.75 (4.07)	9.17 (2.21)	0.496
**SSF (mm)**	7.14 (3.40)	7.58 (3.77)	6.63 (2.91)	0.276
**ASF (mm)**	9.97 (4.48)	10.47 (5.45)	9.41 (3.04)	0.340
**CSF (mm)**	10.06 (3.47)	10.25 (4.02)	9.86 (2.76)	0.650
**Self-esteem Score**	13.61 (1.94)	13.56 (2.06)	13.67 (1.82)	0.818
**Current silhouette**	2.15 (1.23)	1.93 (1.29)	2.39 (1.13)	0.151
**Ideal silhouette**	2.10 (1.21)	1.93 (1.11)	2.28 (1.32)	0.278
**Body Image Score**	0.05 (1.35)	0.00 (1.27)	0.10 (1.47)	0.765

TSF: triceps skinfold; SSF: subscapular skinfold; ASF: abdominal skinfold, CSF: calf skinfold%: percentage, BMI: body mass index, ±SD: standard deviation.

**Table 3 ijerph-19-14197-t003:** Simple and multiple linear regressions between body image and independent variables of female children and adolescents diagnosed with HIV infection (*n* = 32). Florianópolis—SC, 2015/2016.

	Simple				Multiple				
	β (CI95%)	β st	*p*	R²	β (CI95%)	β st	*p*	VIF	R² Final Model
**Age**			0.722	0.004					0.052
8–10	1.0	1.0			1.0	0.163	0.618		
11–15	0.043(−0.200; 0.286)	0.119			−0.082 (−0.421; 0.256)			2.03	
**Sexual maturation**			0.452	0.018					
Prepubescent	1.0	1.0			1.0	0.843	0.224		
Pubescent	0.474 (−0.797; 1.745)	0.622			1.055 (−0.695; 2.804)			1.96	
**Economic Level**			0.256	0.042					
High purchasing power	1.0	1.0			-	-	-	-	-
Low purchasing power	−0.711 (−1.967; 0.545)	0.615			-	-	-	-	-
**ART use**			0.641	0.004		1.199	0.089		
Yes	−0.368 (−1.961; 1.225)	0.780			−2.129 (−4.617; 0.359)			2.55	
No	1.0	1.0			1.0				
**CD4 lymphocytes (cells·mm^−3^)**	0.020 (−0.097; 0.138)	0.025	0.719	0.004	0.055 (−0.0217; 0.131)	0.036	0.151	2.23	
**CD8 lymphocytes (cells·mm^−3^)**	−0.0002 (−0.001; 0.001)	0.0004	0.605	0.009	-	-	-	-	-
**Viral Load (log)**	−0.020 (−0.491; 0.450)	0.230	0.937	0.0002	−0.144 (−0.783; 0.496)	0.308	0.615	1.95	
**Physical activity**			0.182	0.058					
Physically Active	1.0	1.0			-	-	-	-	-
Little Physically Active	−0.731 (−1.825; 0.363)	0.536			-	-	-	-	-
**Television Time (week)**			0.452	0.018					
≥2 h	−0.474 (−1.745; 0.797)	0.623			−	-	-	-	-
<2 h	1.0	1.0			-	-	-	-	-
**Television time (weekend)**			0.785	0.002					
≥2 h	−0.125 (−1.058; 0.807)	0.457			-	-	-	-	-
<2 h	1.0	1.0			-	-	-	-	-
**Computer and Video Game Time (week)**			0.753	0.003					
≥2 h	−0.167 (−1.241; 0.908)	0.526			-	-	-	-	-
<2 h	1.0	1.0			-	-	-	-	-
**Computer and Video Game Time** **(weekend)**			0.127	0.075					
≥2 h	−0.948 (−2.182; 0.286)	0.604			-	-	-	-	-
<2 h	1.0	1.0			-	-	-	-	-
**Lipoatrophy**			0.509	0.014					
Present	−0.365 (−1.485; 0.753)	0.548			-	-	-	-	-
Absent	1.0	1.0			-	-	-	-	-
**BMI (kg·m^−2^)**	9.686 (−14.994–6.931)	0.782	0.902	0.051	-	-	-	-	-
**Body fat (%)**	−0.023 (−0.094; 0.048)	0.034	0.514	0.014	-	-	-	-	-
**TSF (mm)**	−0.009 (−0.126; 0.107)	0.057	0.865	0.001	-	-	-	-	-
**SSF (mm)**	−0.047 (−0.171; 0.078)	0.061	0.450	0.019	−0.236 (−0.457; −0.016)	0.106	**0.036 ***	3.26	
**ASF (mm)**	0.012 (−0.075; 0.098)	0.042	0.785	0.002	-	-	-	-	-
**CSF (mm)**	0.020 (−0.09; 0.13)	0.057	0.725	0.004	0.231 (0.001; 0.462)	0.111	**0.049 ***	4.5	
**Self-esteem Score**	−0.076 (−0.303; 0.152)	0.111	0.502	0.015	−0.231 (−0.511; 0.049)	0.135	0.101	1.57	

TSF: triceps skinfold; SSF: subscapular skinfold; ASF: abdominal skinfold, CSF: calf skinfold; VIF: multicollinearity diagnosis; st: standardized; CI: confidence interval; *p*: *p*-value; *: *p* < 0.05.

**Table 4 ijerph-19-14197-t004:** Simple and multiple linear regressions between body image and independent variables of male children and adolescents diagnosed with HIV infection (*n* = 28). Florianópolis—SC, 2015/2016.

	Simple				Multiple				
	β (CI95%)	β st	*p*	R²	β (CI95%)	β st	*p*	VIF	R² Final Model
**Age**			0.122	0.089					0.724
8–10	1.0	1.0			-	-	-	-	
11–15	−1.000 (−2.286; 0.286)	0.626			-	-	-	-	
**Sexual maturation**			**0.040 ***	0.146			**<0.01 ***	1.82	
Prepubescent	1.0	1.0			1.0	1.0			
Pubescent	−1.348 (−2.663; −0.034)	0.639			−1.595 (−2.615; −0.5751)	0.481			
**Economic Level**							**0.03 ***	1.58	
High purchasing power	1.0	1.0			1.0	1.0			
Low purchasing power	1.153 (−2.150; 2.381)	1.102	0.100	0.0004	−1.671 (−3.187; −0.155)	0.715			
**ART use**									
Yes	0.905 (−0.393; 2.202)	0.631	0.163	0.073	-	-	-	-	-
No	1.0	1.0			-	-	-	-	-
**CD4 lymphocytes (cells·mm^−3^)**	0.007 (−0.058; 0.073)	0.032	0.820	0.002	−0.119 (−0.182; −0.057)	0.029	**<0.01 ***	3.19	
**CD8 lymphocytes (cells·mm^−3^)**	−0.0002 (−0.0012; 0.0007)	0.0004	0.641	0.008	-	-	-	-	-
**Viral Load (log)**	−0.555 (−1.177; 0.068)	0.303	0.070	0.114	−1.489 (−2.102; −0.875)	0.289	**<0.01 ***	3.04	
**Physical activity**			0.577	0.012			**0.03 ***	3.20	
Physically Active	1.0	1.0			-	-	-	-	-
Little Physically Active	0.326 (−0.861; 1.514)	0.578			0.824 (0.079; 1.569)	0.351			
**Television Time (week)**									
≥2 h	0.649 (−0.573; 1.871)	0.594	0.284	0.043	-	-	-	-	-
<2 h	1.0	1.0	1.0	1.0	-	-	-	-	-
**Television time (weekend)**									
≥2 h	0.322 (−0.889; 1.533)	0.589	0.589	0.011	-	-	-	-	-
<2 h	1.0	1.0	1.0	1.0	-	-	-	-	-
**Computer and Video Game Time (week)**							**<0.01 ***	2.32	
≥2 h	−0.631 (−1.772; 0.511)	0.556	0.260	0.047	−4.037 (−5.288; −2.785)	0.590			
<2 h	1.0	1.0	1.0	1.0	1.0	1.0			
**Computer and Video Game Time (weekend)**							**<0.01 ***	4.05	
≥2 h	−0.056 (−1.226; 1.113)	0.569	0.922	0.0004	2.023 (0.811; 3.234)	0.571			
**Lipoatrophy**									
Present	-	-	-	-	-	-	-	-	-
Absent	-	-	-	-	-	-	-	-	-
**BMI (kg·m^−2^)**	−12.810 (−37.204; 11.584)	0.119	0.290	0.042	-	-	-	-	-
**Body fat (%)**	0.001 (−0.079; 0.082)	0.039	0.970	0.001	−0.048 (−0.100; 0.003)	0.024	0.06	3.79	
**TSF (mm)**	−0.042 (−0.310; 0.226)	0.131	0.750	0.003	-	-	-	-	-
**SSF (mm)**	−0.006 (−0.210; 0.197)	0.099	0.949	0.0001	-	-	-	-	-
**ASF (mm)**	−0.078 (−0.271; 0.114)	0.094	0.410	0.026	0.043 (−0.080; 0.165)	0.058	0.470	1.40	
**CSF (mm)**	0.039 (−0.175; 0.253)	0.104	0.712	0.005	-	-	-	-	-
**Self-esteem Score**	0.067 (−0.391; 0.257)	0.157	0.675	0.007	−0.109 (−0.319; 0.101)	0.099	0.28	1.46	

TSF: triceps skinfold; SSF: subscapular skinfold; ASF: abdominal skinfold, CSF: calf skinfold; VIF: multicollinearity diagnosis; st: standardized; CI: confidence interval; *p*: *p*-value; * and bold: *p* < 0.05.

## Data Availability

Not applicable.

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
