# Peer review of "Factors Correlated with Body Image Dissatisfaction in Children and Adolescents Diagnosed with HIV: A Cross-Sectional Study"

_ijerph, 2022, doi:10.3390/ijerph192114197_

Round 1

Reviewer 1 Report

Thank you for your research and submission. In addition, I would like to make some recommendations.

1. I recommend replacing the hiv mentioned in the title with a capital letter.

2. This is my personal opinion, but I think visualizing and presenting the model of the study is one of the good ways to help readers understand. Please consider it.

3. I think it is necessary to modify the form of the reference.

4. Unnecessary parts should be deleted at the end of the reference.

Author Response

Good morning, we thank you for the careful review of the article and we are attaching our response letter to the reviewer.

Reviewer 2 Report

As the title suggests the manuscript describes body image dissatisfaction in children and adolescents with HIV (based on silhouette scale). The paper was well written, number of subjects was good, methods and statistical analyses seem appropriate.

Introduction: Well written. One suggestion perhaps a few more sentences on ART’s adverse effects on physical appearance specifically on psychological and behavioral issues.

The introduction – lines 78-81- note importance to educators and health care professionals, but no follow up in how results can be used by them in discussion.

 Discussion: Suggest a more robust discussion on what does this mean for educators and health care professionals, for the children. Are the results similar to those found in other studies on non-hiv and hiv children/ adolescents, not just rehash of the results. What can educator/ health professional do with the info gained from this study.

Author Response

(The authors gave the same response as above.)

Reviewer 3 Report

The article is very well written and presented. The topic is interesting and could have important practical applications. The material and methods part is very accurate and punctual in the explanations.

However, there are some point to fix. 

In the abstract, there is not a background about the topic. Please add something before starting with the aim of the article. 

I think that the introduction part could be improved; it seems a little bit consufing.  Please, follow a logical thread, from a topic to another.

I suggest you this article, in which authors explain about two index that could be interesting for your article and research. 

"Body perception and body composition: assessment of perception inconsistency by a new index", by Zaccagni L.; Rinaldo N.; Bramanti B.; Mongillo J.; Gualdi-Russo E. J Transl Med, 2020, 14, 18(1): 201, doi: 10.1186/s12967-019-02201-1. 

Please homogeineized the use of stitches and commas in Table 3 and table 4. In Table 3, why did not think to use three economical levels?

Discussion, I would report the objectives of the article before immediately reporting the results. In addition, the part of the discussions is horizontal rather than vertical. 

Author Response

(The authors gave the same response as above.)

Round 2

Reviewer 3 Report

Dear Authors, 

Thank you for following the suggestions and implementing the article. Now. for me, it is ready for the publication.